# Identification, Characterization, and Functional Analysis of Chitin Synthase Genes in *Glyphodes pyloalis* Walker (Lepidoptera: Pyralidae)

**DOI:** 10.3390/ijms21134656

**Published:** 2020-06-30

**Authors:** Zuo-Min Shao, Yi-Jiangcheng Li, Jian-Hao Ding, Zhi-Xiang Liu, Xiao-Rui Zhang, Jun Wang, Sheng Sheng, Fu-An Wu

**Affiliations:** 1Jiangsu Key Laboratory of Sericultural Biology and Biotechnology, School of Biotechnology, Jiangsu University of Science and Technology, Zhenjiang 212018, China; shaozuomin23@163.com (Z.-M.S.); 192310019@stu.just.edu.cn (Y.-J.L.); lostone1@126.com (J.-H.D.); 18852864112@163.com (Z.-X.L.); zxr1131286377@sina.com (X.-R.Z.); wangjun@just.edu.cn (J.W.); parasitoids@163.com (S.S.); 2Key Laboratory of Silkworm and Mulberry Genetic Improvement, Ministry of Agriculture and Rural Affairs, Sericultural Research Institute, Chinese Academy of Agricultural Science, Zhenjiang 212018, China

**Keywords:** *Glyphodes pyloalis* Walker, chitin synthase, DFB, RNA interference, pest control

## Abstract

*Glyphodes pyloalis* Walker (*G. pyloalis*) causes significant damage to mulberry every year, and we currently lack effective and environmentally friendly ways to control the pest. Chitin synthase (CHS) is a critical regulatory enzyme related to chitin biosynthesis, which plays a vital role in the growth and development of insects. The function of CHS in *G. pyloalis*, however, has not been studied. In this study, two chitin synthase genes (*GpCHSA* and *GpCHSB*) were screened from our previously created transcriptome database. The complete coding sequences of the two genes are 5,955 bp and 5,896 bp, respectively. Expression of *GpCHSA* and *GpCHSB* could be detected throughout all developmental stages. Relatively high expression levels of *GpCHSA* occurred in the head and integument and *GpCHSB* was most highly expressed in the midgut. Moreover, silencing of *GpCHSA* and *GpCHSB* using dsRNA reduced expression of downstream chitin metabolism pathway genes and resulted in abnormal development and wings stretching, but did not affect normal pupating of larvae. Furthermore, the inhibitor of chitin synthesis diflubenzuron (DFB) was used to further validate the RNAi result. DFB treatment significantly improved expression of *GpCHSA*, except *GpCHSB*, and their downstream genes, and also effected *G. Pyloali* molting at 48 h (62% mortality rate) and 72 h (90% mortality rate), respectively. These results show that *GpCHSA* and *GpCHSB* play critical roles in the development and wing stretching in *G. pyloalis* adults, indicating that the genes are attractive potential pest control targets.

## 1. Introduction

*Glyphodes pyloalis* Walker is an important mulberry pest which is widely distributed throughout major mulberry growing areas of China, India, Korea, Japan, Pakistan, and Burma. This insect damages sericulture not only by feeding on mulberry but also by transmitting viruses to the silkworm [1]. In recent years, the outbreak of *G. pyloalis* has resulted in severe losses to local mulberry farmers. Currently, using conventional insecticides is the principal means to control *G. pyloalis* populations. However, the heavy use of insecticides in mulberry growing regions has caused silkworm poisoning and environmental pollution, as well as increased *G. pyloalis* resistance to a variety of chemical insecticides including organophosphates, pyrethroids, and carbamates [2]. Therefore, there is an urgent need for environmentally-friendly methods to control *G. pyloalis*.

Chitin is the second most abundant natural polymer (after cellulose), and it is a polymer of *N*-acetyl-*β*-d-glucosamine (GlcNAc). It is widely distributed throughout fungus, arthropod, nematode, and marine organism [3,4,5,6]. In insects, chitin is an essential component of the cuticle and peritrophic matrix (PM), which plays an important role in keeping insect shape and protecting itself from external stresses, respectively. Periodic molting is the process that involves the shedding and replacing of the thick insect exoskeleton, and is an essential step that occurs during insect growing development. Throughout the process, part of the old cuticle is digested, and then new chitin is synthesized. Chitin metabolism is regulated by three main enzymes, chitin synthase (CHS), chitin deacetylase (CDA), and chitinase (CHT) [7,8]. CHS was first identified in yeast and fungi [9,10] and contains two conserved motifs, EDR and QRRRW [11], which are found in CHS of all chitin-synthesizing organisms [12]. It is one kind of transmembrane protein from a large glycosyltransferases family, and is involved in catalyzing the polymerization of UDP-*N*-acetylglucosamine (UDP-GlcNAc) from chitin. *CHS* have been found existed in many insect species [13], including *Drosophila melanogaster* (*D. melanogaster*)[14], *Choristoneura fumiferana* [15], *Anopheles gambiae* (*A. gambiae*)[16], *Locusta migratoria manilensis* [17], *Tribolium castaneum* (*T. castaneum*)[18], *Plutella xylostella* [19], and *Manduca sexta* [20]. Currently, two kinds of *CHS* genes in insects have been identified, *CHSA* and *CHSB* (also called *CHS1* and *CHS2*). *CHSA* is mainly reported in the cuticle and tracheae, and *CHSB* is mainly reported in midgut epithelial cells and associated with the formation of the PM. Therefore, CHS plays an important role in insect growing development by regulating molting and cuticle regeneration. Disruption of chitin synthesis has been extensively recognized as a promising means to control insect populations [12,21]. 

Diflubenzuron (DFB) [1-(4-chlorophenyl)-3-(2.6-diflubenzoyl)urea] is the first benzoylphenylurea (BPU) chitin synthesis inhibitor that acts by controlling chitin synthesis in various pest species [21,22]. Exposure of *Panonychus citri* to DFB causes abortive molting and high rates of larvae mortality [23]. Treatment of *Anopheles quadrimaculatus* (*A. quadrimaculatus*) with DFB significantly reduces chitin content. Moreover, the expression of *AqCHS1* was significantly upregulated after exposure to DFB in *A. quadrimaculatus* [12]. RNA interference (RNAi) is a useful tool used to analyze gene function in different organisms by delivering gene-specific, double-stranded RNA (dsRNA). It is also known as a promising approach for pest control [24]. RNAi has been reported to assess the function of *CHS* in many insect species. Silencing of *A. gambiae CHS1* and *CHS2* via chitosan/*AgCHS* dsRNA-based nanoparticles feeding resulted in the downregulation of both genes and reduced chitin accumulation. The treatment lead to enhanced larval susceptibility to diflubenzuron [25]. Silencing of *Diaphorina citri CHS* caused malformed phenotypes, increased mortality, and decreased molting rates [26]. The function of *Acyrthosiphon pisum CHS* was studied by injecting insects with dsRNA, and results revealed that *CHS* plays an important function in embryonic development and nymphal growth [24]. RNAi was used to target *Toxoptera citricida CHS* via feeding fourth instar with dsRNA, which rendered insects incapable of molting [27]. However, the function of *CHS* in *G. pyloalis* remains unclear. Thus, clarification of the function of *CHS* genes in *G. pyloalis* may contribute to identify candidate molecular targets for control of the pest.

In this study, *GpCHSA* and *GpCHSB* were identified from previous transcriptome data [28]. The sequences of the genes were analyzed using bioinformatics, and expression patterns of the genes in different tissues and developmental stages were analyzed using RT-qPCR. The function of each gene was studied using DFB and dsRNA treatment. Our study preliminarily clarifies the function of *GpCHSA* and *GpCHSB* in the chitin metabolism pathway and may facilitate the finding of new tools for managing *G. pyloalis*.

## 2. Results

### 2.1. Characterization of the GpCHSA and GpCHSB Sequences

*GpCHSA* and *GpCHSB* cDNA sequences have been uploaded to NCBI GenBank, and accession numbers are MN915086 and MN915087, respectively. The full-length of *GpCHSA* is 5,955 bp, which contains a 4,695 bp open reading frame (ORF) that encodes a 1,564 amino acid protein with a predicted molecular mass (MW) of 178.52 kDa and isoelectric point (pI) of 6.54. Full-length of *GpCHSB* is 5,896 bp, and contains a 4,590 bp ORF that encodes a protein with 1,529 amino acid residues. The protein has a predicted MW of 174.99 kDa and pI of 6.27. The putative catalytic domain of GpCHSs included two representative motifs EDR and QRRRW (Figure 1). The catalytic functional domain of chitin synthases shared 94.02% amino acid identity in different Lepidoptera insects based on the multiple sequence alignment (Figure 1). Moreover, the full structure of the GpCHSA protein consists of 13 transmembrane regions, one low complexity region, and 2 coiled-coil regions (Figure 2A). GpCHSB protein also contains 13 transmembrane regions, two low complexity regions, and one coiled coil regions (Figure 2B). Detailed sequence information of GpCHSA and GpCHSB are showed in Appendix A.

To get the phylogenetic tree between CHSA and CHSB in different species, MEGA-X software was used. A total of 33 CHSs from 20 species were grouped into two phylogenetic groups, CHSA and CHSB, indicating that some kinds of difference between of them do exist. Moreover, the result showed that both GpCHSA and GpCHSB have a close relationship with *Cnaphalocrocis medinalis* CHSA and CHSB, respectively (Figure 3).

### 2.2. Spatio-Temporal Expression Patterns of GpCHSA and GpCHSB

To preliminarily get the specific biological function of *GpCHSA* and *GpCHSB*, relative expression levels of both genes were detected in different developmental stages and tissues of fifth instar larvae using RT-qPCR. The expression levels showed that both genes expressed throughout all of developmental stages, which indicated that *GpCHSA* and *GpCHSB* were involved in chitin metabolism throughout *G. pyloalis* whole life. Moreover, *GpCHSA* expression was determined to be the highest in the pupa, and *GpCHSB* expression peaked at the larval stage (Figure 4). Moreover, the expression levels of *GpCHSA* in the head and integument were 272 and 108 fold greater than that in the midgut, respectively. The greatest expression level of *GpCHSB* was in the midgut that was 727 times higher than that in the integument (Figure 4). Therefore, we suspected that *GpCHSA* might participate in the integument chitin synthesis pathway, while *GpCHSB* is more likely to be involved in the midgut chitin synthesis pathway.

### 2.3. Inhibition of GpCHSA Expression Could Downregulate Expression of Its Downstream Genes

To further explore the function of *GpCHSA* in the integument chitin synthesis pathway, dsRNAs were injected into larvae at the first day of fifth instar stage. The expression level of *GpCHSA* was measured at 24, 48, and 72 h after injection with dsCHSA using RT-qPCR. Results revealed that the expression of *GpCHSA* was significantly downregulated at 48 h after injection as compared with the control group (Figure 5). Moreover, the expression of *GpCDA1*, *GpCDA2*, and *GpCHT3a* showed similarly affected expression patterns with *GpCHSA* after knockdown of *GpCHSA*. Briefly, expression of *GpCHSA* was significantly upregulated at 24 h and downregulated at 48 h after injection with dsRNA (Figure 5), indicating *GpCHSA* plays a vital role in regulating *GpCDA1*, *GpCDA2*, and *GpCHT3a* expression. 

### 2.4. RNAi of GpCHSB Resulted in Significant Downregulation of Its Downstream Genes

The method used to assess RNAi effect of *GpCHSB* was the same as was used to assess effects of *GpCHSA* knockdown described above. Expression level of *GpCHSB* was measured at 24, 48, and 72 h after injection with dsCHSB using RT-qPCR. Results revealed that *GpCHSB* expression was significantly downregulated at 48 h after injection with dsRNA (Figure 6). Moreover, the expression levels of *GpCDA5*, *GpCHT3b*, and *GpCHT-h* were significantly downregulated at 48 h after knockdown of *GpCHSB* (Figure 6), while *GpCDA5* and *GpCHT-h* were upregulated at 72 h post-injection. 

### 2.5. RNAi of GpCHSA and GpCHSB Resulted in an Adverse Effect on *G. pyloalis* Development

To further confirm the functions of *GpCHSA* and *GpCHSB*, adult and pupal phenotypes of *G. pyloalis* were assessed after injection with dsCHSA and dsCHSB. Results showed that knockdown of *GpCHSA* and *GpCHSB* adversely affected *G. pyloalis* wing development, including abnormal wing stretching and partial loss (Figure 7). However, pupation of fifth instar larvae was not affected by dsRNA injection (Figure 7). 

### 2.6. DFB Treatment Could Affect Expression of GpCHSA, GpCHSB, and Their Downstream Genes

In order to further validate findings observed after injection with dsCHSA and dsCHSB, fourth instar larvae of *G. pyloalis* after treatment with DFB was assessed at different times. Results revealed that *GpCHSA* expression was significantly upregulated at 24 and 48 h post-exposure to DFB in the integument (Figure 8). Moreover, relative expression levels of *GpCDA1, GpCDA2* and *GpCHT3a* involved in the integument chitin metabolism pathway were also significantly upregulated at 24 h post-treatment with DFB (Figure 8). 

The relative expression level of *GpCHSB* was significantly down-regulated at 12 h post-exposure to DFB in the midgut. Subsequently, expression was upregulated at 24 and 48 h post-exposure with DFB did not significant differ (Figure 8). Moreover, expression levels of *GpCDA5* and *GpCHT-h* were significantly upregulated at 12 h post-exposure to DFB, while *GpCHT3b* was downregulated (Figure 8). After 12 h of DFB exposure, three downstream genes of *GpCHSB* were downregulated. However, the same downstream genes were upregulated at 48 h post-DFB treatment (Figure 8).

### 2.7. DFB Treatment Adversely Affected *G. pyloalis* Development

In order to assess effects of DFB treatment on *G. pyloalis* development and mortality, a leaf-dip bioassay was conducted. Results indicated abnormal molting occurred after treatment with DFB (Figure 9A,B). Moreover, cumulative mortality of *G. pyloalis* larvae was significantly increased at 48 h (62%) and 72 h (90%) after exposure to DFB as compared with control insects (Figure 9C).

## 3. Discussion

As use of pesticides has become widespread, *G. pyloalis* insecticide resistance has become severity in recent years. Hence, it is necessary to find alternate effective and environmentally friendly methods to solve the problem. Chitin is an essential component of the cuticle and PM of *G. pyloalis*, which serves as an initial barrier that protects *G. pyloalis* from pesticides. To identify the genes involved in chitin biosynthetic pathway, the integument, midgut, and whole larvae of *G. pyloalis* were sequenced using RNA-Seq method in our previous study [28]. A total of 19 genes were identified that encode chitin metabolism-related enzymes, and *GpCHSA* and *GpCHSB* related to the chitin biosynthesis are two of them. In this study, to further validate the function of *GpCHSA* and *GpCHSB* in the chitin metabolism pathway, RNAi and the inhibitor of chitin synthesis DFB were used. 

It was reported that many enzymes were involved in the chitin metabolism progress. Chitin synthase (CHS) plays a key role in chitin biosynthesis during insect growth and metamorphosis in Diptera, Coleoptera, Lepidoptera, Hemiptera, and Hymenoptera [8]. In fungal species, there are more than 20 fungal CHS genes have been identified, while only two CHS genes (CHSA and CHSB) occur in insects [4]. *CHSA* is highly expressed in the cuticle and is essential for the formation of integument. *CHSB* is greatly expressed in the midgut and is an important component of the PM [30]. However, CHS genes of *G. pyloali* have not been well studied. In this study, complete coding sequences of *GpCHSA* and *GpCHSB* were identified from our previously constructed *G. pyloali* transcriptome database. *GpCHSA* and *GpCHSB* contain two conserved catalytic domains, EDR and QRRRW, which are involved in chitin synthesis and conserved among different species. This indicated their likely function in *G. pyloali* chitin synthesis (Figure 1). 

GpCHSA and GpCHSB were divided into two different groups via phylogenetic analysis, indicating that the two genes are likely involved in two different chitin metabolism pathways. This finding was further validated by the observation that expression levels of *GpCHSA* were higher in the integument as compared with others, while higher expression of *GpCHSB* in the midgut (Figure 3 and Figure 4). These results are consistent with previous studies involving other species [31,32,33,34]. Moreover, significant differences between expression levels of *GpCHSA* and *GpCHSB* were observed throughout all selected development stages, which indicated that the two *CHS* genes likely played important roles in the whole life of *G. pyloali* (Figure 4). Moreover, enhanced expression of *GpCHSA* in the pupa and relatively high expression levels of *GpCHSB* throughout larval stages validated their roles in chitin metabolism pathways of the integument and midgut, respectively (Figure 4). 

RNAi has been proved to be an efficient molecular biology technique, and has been widely used to study gene function in insects. RNAi also has been considered as a potential mean to control pests [35]. Since the chitin synthesis pathway does not exist in vertebrates and plants, RNAi targeting of chitin synthases is an attractive potential means to control insect populations [36]. It has been widely reported that knockdown of *CHS* genes using dsRNA is feasible in lepidopteran insects. For example, feeding of *Helicoverpa armigera* larvae with *HaCHSA* dsRNA resulted in reduced body weight, growth and pupation rates [37]. Knockdown of *SeCHSA* in *Spodoptera exigua* larvae using dsRNA reduced survival rates in larvae, prepupa and pupa [38]. In this study, larvae were injected with dsRNA on the first day of fifth instar stage to study the function of *GpCHSA* and *GpCHSB* in the chitin metabolic pathway. Significant downregulation of *GpCHSA* and *GpCHSB* at 48 h post-injection indicated that use of dsRNA was effective (Figure 5 and Figure 6), and the significant decrease of the expression of downstream genes of *GpCHSA* and *GpCHSB*, including *GpCDA1, GpCDA2*, *GpCHT3a* and *GpCDA5, GpCHT-h*, *GpCHT3b*, respectively, indicated *GpCHSA* and *GpCHSB* played vital roles in the regulation of their expression. Moreover, we found that *G. pyloali* wing development was affected by dsCHSA and dsCHSB (Figure 7), which was consistent with the results of other species that also occurred abnormal development after knockdown of *CHS* expression [17,39].

DFB as a chitin synthesis inhibitor has been recommended for spraying forage crops in order to protect against several pests [21,40]. DFB is effective because it causes abnormal procuticle deposition and abortive molting in pests [12,23,41]. Its function mainly reduces chitin content in insects, and frequently leads to the upregulation of CHSA. This may be caused by a feedback regulatory mechanism that attempts to compensate for decreased enzyme content [12,30]. In this study, *GpCHSA* was significantly upregulated approximately 2 fold at 24 h after exposure to DFB (Figure 8), which was consistent with previous reports [12,23]. However, levels of *GpCHSB* were not significantly altered (Figure 8), which reflected that *GpCHSA* might play a more important role in the response to DFB treatment than *GpCHSB*. It has been shown previously that DFB affects species differently. For example, no effects of DFB were observed in *D. melanogaster* [42] and *T. castaneum* [43]. Moreover, downstream genes of *GpCHSA*, including *GpCDA1, GpCDA2* and *GpCHT3a*, increased post-upregulation of *GpCHSA* expression (Figure 8). This further validated the results of RNAi, and indicated that *GpCHSA* was involved in regulating expression of its downstream genes. Though the expression of *GpCHSB* was not significantly altered by DFB treatment, but the same trend of the downstream genes of *GpCHSB* with itself at 24 and 48 h after DFB treatment (Figure 8), including *GpCDA5, GpCHT-h,* and *GpCHT3b*, indicated their close relationship with *GpCHSB*. Furthermore, larvae fed DFB displayed abnormal molting and increased mortality as compared with control insects (Figure 9), indicating that DFB treatment affected *G. pyloali* development by disturbing the chitin metabolism pathway. These results further suggest the role of *GpCHSA* and *GpCHSB* in the chitin metabolism pathway. 

Based on results described above, it was reasonable to conclude that *GpCHSA* and *GpCHSB* were involved in *G. pyloali* cuticle and PM chitin metabolism, respectively*. GpCHSA* regulated expression of its downstream genes *GpCDA1*, *GpCDA2,* and *GpCHT3a*. *GpCDA1* and *GpCDA2* participates in cuticle chitin deacetylation, and *GpCHT3a* in cuticle chitin degradation (Figure 10). Moreover, *GpCHSB* could regulate the expression of its downstream genes *GpCDA5*, *GpCHT-h* and *GpCHT3b*. *GpCDA5* regulated PM chitin deacetylation, and *GpCHT-h* and *GpCHT3b* affected chitin degradation in the PM (Figure 10). Furthermore, *GpCHSA* and *GpCHSB* played a critical role molting and wing development in *G. pyloalis* via regulation of chitin metabolism. Taken together, these findings indicate that *GpCHSA* and *GpCHSB* may be attractive new targets for *G. pyloalis* control. 

## 4. Materials and Methods

### 4.1. Bioinformatics Analysis 

ORF finder (https://www.ncbi.nlm.nih.gov/orfnder/) was used to predicte the open reading frames (ORFs) of putative GpCHSA and GpCHSB genes. ExPASy (https://web.expasy.org/compute_pi/) was used to predicte the theoretical isoelectric point (pI) and molecular weight of each enzyme. DNAMAN 8.0 software (Lynnon Corporation, Quebec, Canada) was used to perform the multiple alignments of various protein sequences. Conserved motifs were predicted by using SMART software (http://smart.embl-heidelberg.de/). Phylogenetic analysis was conducted using MEGA-X software and the neighbor-joining method with 1,000 bootstrap replications. The protein sequences of GpCHS homologs of 19 other species were acquired from GenBank (http://www.ncbi.nlm.nih.gov/). GenBank IDs of each protein are listed in Appendix A. 

### 4.2. Insect Rearing and Sample Preparation

*G. pyloalis* were maintained in the Key Laboratory of Silkworm and Mulberry Genetic Improvement, Ministry of Agriculture, Sericultural Research Institute, Chinese Academy of Agricultural Science, Zhenjiang, Jiangsu, China. The larvae were reared in an insect-rearing room with fresh mulberry leaves. The rear conditions are 25 ± 1 °C, 60–80% relative humidity, and a 14 h light and 10 h dark photoperiod. 

The different development stages of *G. pyloalis* has been classified based on insect morphological features. Ten samples were mixed to minimize individual genetic differences in detecting the relative expression level in different life stages. Different tissues of 50 larvae in fifth instar were dissected for detecting relative expression levels in different tissues. All samples were kept in −80 °C for further use. Triple replicates of each gene in these samples were performed.

### 4.3. Total RNA Extraction and cDNA Synthesis

TRIzol reagent (Invitrogen, New York, NY, USA) was used to extract the total RNA in accordance with the manufacturer’s protocols. The NanoDrop 2000 spectrophotometer (Thermo Fisher Scientific, New York, NY, USA) was used to perform the RNA quantification. RNA purity was determined by assessing optical density (OD) absorbance ratios at OD_260/280_ and OD_260/230_. The integrity of RNA was analyzed via 1% agarose gel electrophoresis with ethidium bromide staining. Reverse transcription was performed using a PrimeScript^™^ RT reagent Kit with a gDNA Eraser (TaKaRa Biotechnology Co. Ltd., Dalian, China). Briefly, reaction was incubated at 37 °C for 15 min and then 85 °C for 5 s. Synthesized cDNA was preserved at −20 °C for further use.

### 4.4. Quantitative Reverse Transcription PCR (RT-qPCR)

NCBI Primer-BLAST software (https://www.ncbi.nlm.nih.gov/tools/primer-blast/) was used to design the specific primers that were used for RT-qPCR analysis and shown in Appendix A. The reaction was analyzed using a QuantStudio™ Real-Time PCR system (Thermo Fisher Scientific, Applied Biosystems, New York, NY, USA). Each reaction used 15 μL reaction system containing 1.5 μL cDNA, 7.5 μL TB Green Fast qPCR Mix (TaKaRa Biotechnology Co. Ltd., Dalian, China), 0.6 μL of each gene-specific primer (0.4 μM), 0.3 uL of ROX Reference Dye II and 4.5 μL of RNase free H_2_O. *G. pyloali ribosomal protein L32* (*GpRpl32*) was used as reference standard gene [44]. The 2^−ΔΔ*C*t^ method was used to calculate the relative expression level of each gene. Statistical analysis was conducted using R version 4.0.0. The triplicate data of different groups were analyzed via One-way ANOVA with Tukey’s posttest.

### 4.5. dsRNA Synthesis and Injection

To ensure the RNAi efficiency, two specific targets of the functional domain of *GpCHSA* and *GpCHSB* genes were designed. The specific target of green fluorescent protein (GFP) was selected as negative treatment. Primers used to synthesize dsRNA by Sangon Biotechnology (Shanghai, China) are listed in Appendix A. The in vitro Transcription T7 Kit (for dsRNA synthesis; TaKaRa Biotechnology Co. Ltd., Dalian, China) was used to synthesize dsCHSA, dsCHSB, and dsGFP according to the manufacturer’s instructions. The integrity of the dsRNA was determined by 3% agarose gel electrophoresis. The quality and concentration of purified dsRNA were measured using a NanoDrop 2000 spectrophotometer (Thermo Fisher Scientific, NY, USA). Newly synthesized dsRNA was saved in −80 °C for further use.

dsCHSA and dsCHSB were dissolved in DEPC water to a final concentration of 2.0 μg/μL. Equal volumes of the two targets were mixed, and 1.0 μL of the mixture dsRNAs were injected into each larva on the first day of fifth instar development period using microneedles. Thirty (30) larvae were under dsRNA and ddH_2_O treatment in each group. Expression levels of the two genes and their downstream genes in the integument and midgut were analyzed using RT-qPCR after RNAi at 24, 48, and 72 h post-injection, respectively. Triple biological replicates were done for each group. The method used to investigate phenotypes of *G. pyloalis* in different stages after RNAi was the same with sample preparation as descripted above.

### 4.6. Leaf Dip Bioassay

To determine the effect of DFB on *GpCHSA* and *GpCHSB* and their downstream genes expression, the mortality rates of fourth instar larvae of *G. pyloalis* were determined using a leaf-dip bioassay using a previous protocol with some modifications [23]. Briefly, mulberry leaves 9-cm in diameter were dipped in the final concentration 500 μg/mL of DFB (LKT Laboratories Inc., Saint Paul, MN, USA) that was dissolved in 0.1% Triton-100 for 3 min and air dried. Leaves treated with 0.1% Triton-100 were used as controls. Thirty of fourth instar larvae exposed to DFB and Triton-100 were kept in the insect-rearing room for 24, 48, and 72 h to investigate phenotypic effects of treatment and cumulative mortality, respectively. Surviving larvae were collected for further analysis. The experiment was performed using triple biological replicates.

## Figures and Tables

**Figure 1 ijms-21-04656-f001:**
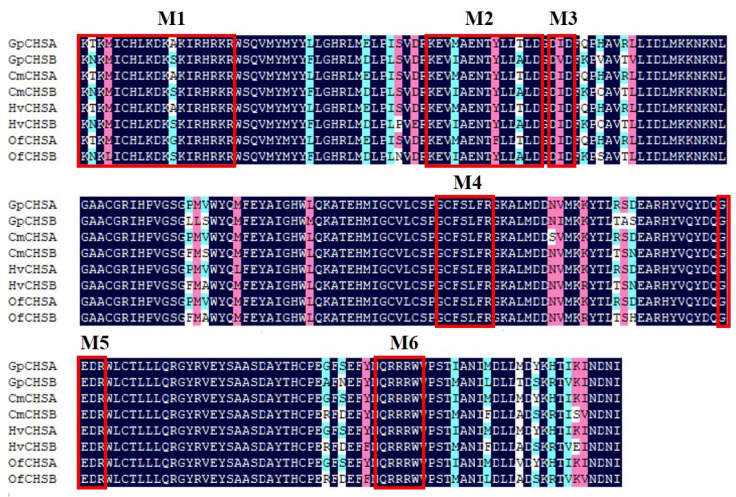
The analysis of putative catalytic functional domains of chitin synthases (CHSs) among different Lepidoptera insects based on the multiple sequence alignment. Dark blue represents identical amino acids. Pink and aqua represent positively charged amino acids. Non-conserved positions have a white background. Six characteristic motifs M1-6 of highly conserved regions of glycosyltransferases (family 2) enzymes were shown in red boxes [22,29]. Cm, *Cnaphalocrocis medinalis*, CmCHSA, AJG44538.1, CmCHSB, AJG44539.1; Hv, *Heortia vitessoides*, HvCHSA, AZQ19982.1, HvCHSB, AZQ19981.1; Of, *Ostrinia furnacalis*, OfCHSA, ACF53745.1, OfCHSB, ABB97082.1.

**Figure 2 ijms-21-04656-f002:**
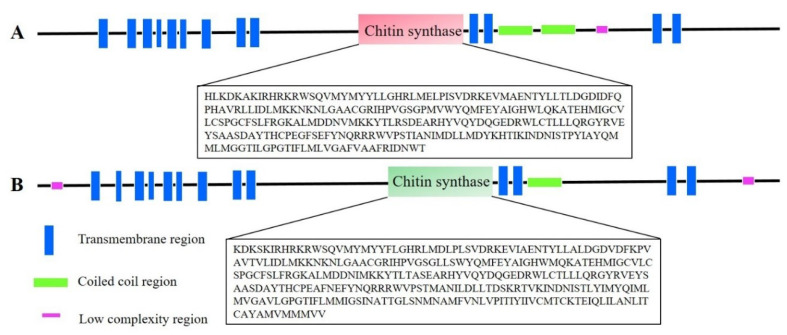
The predicted conserved functional domains of GpCHSA and GpCHSB. Different colors are used to mark predicted domains. The conserved functional domain of GpCHSA consists of 261 amino acids (**A**), and GpCHSB has 302 amino acids (**B**). Conserved sequences were identified using SMART (http://smart.embl-heidelberg.de/) website and indicated in boxed.

**Figure 3 ijms-21-04656-f003:**
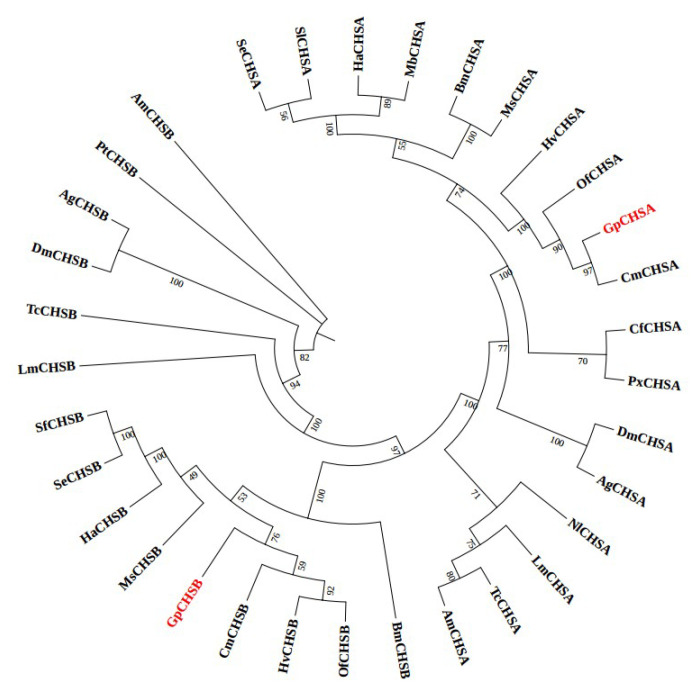
Phylogenetic relationship analysis of CHSs in different identified insect species using the neighbor-joining method with 1,000 bootstraps. Bm, *Bombyx mori*; Cf, *Choristoneura fumiferana*; Cm, *Cnaphalocrocis medinalis*; Dm, *Drosophila melanogaster*; Ha, *Helicoverpa armigera*; Lm, *Locusta migratoria*; Ms, *Manduca sexta*; Px, *Plutella xylostella*; Se, *Spodoptera exigua*; Tc, *Tribolium castaneum*; Of, *Ostrinia furnacalis*; Sf, *Spodoptera frugiperda*; Hv, *Heortia vitessoides*; Ag, *Anopheles gambiae*; Pt, *Parasteatoda tepidariorum*; Am, *Apis mellifera*; Mb, *Mamestra brassicae*; Nl, *Nilaparvata lugens;* Sl, *Spodoptera litura*.

**Figure 4 ijms-21-04656-f004:**
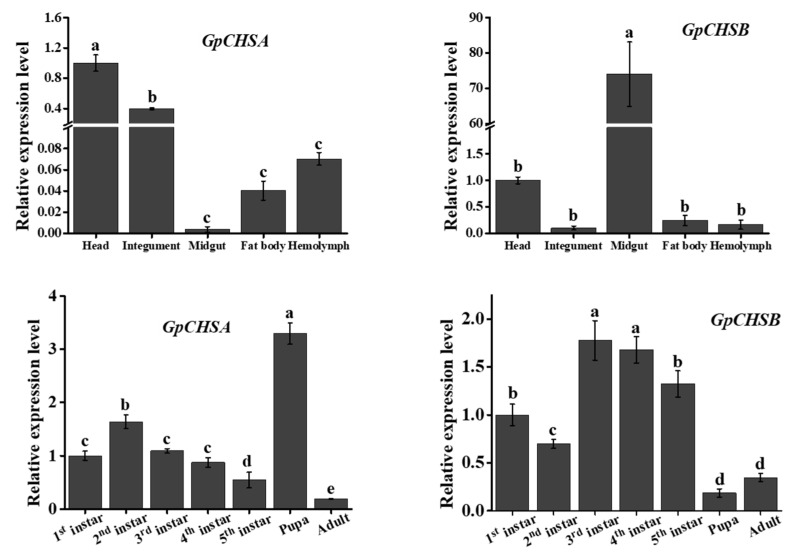
Analysis of *GpCHSA* and *GpCHSB* expression in different developmental stages and tissues of fifth instar larvae using RT-qPCR. *GpRpl32* was used to normalize the data that are showed as mean ±standard error, the mean is the triple independent repeats. The 2^−ΔΔ*C*t^ method was adopted to calculate the relative expression level. Differences among triple repeats were analyzed using one-way analysis of variance (Systat, Inc., Evanston, IL) with Tukey’s post-hoc test using R version 4.0.0. Different letters (a, b, c, d, e) represent significant difference (*p* < 0.05).

**Figure 5 ijms-21-04656-f005:**
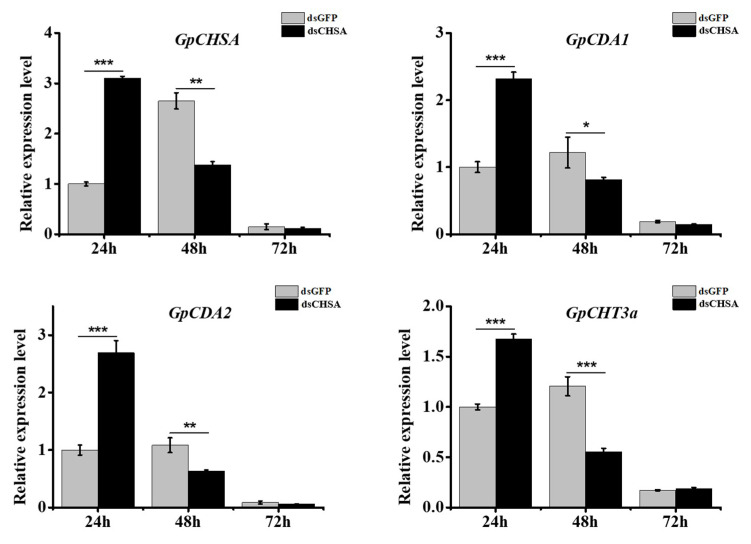
The analysis of expression patterns of *GpCHSA* and its downstream genes post-injection with dsCHSA in the integument at different times using RT-qPCR. *GpRpl32* was used to normalize the data that are showed as mean ±standard error, the mean is the triple independent repeats. The 2^−ΔΔ*C*t^ method was adopted to calculate the relative expression level. Differences among triple repeats were analyzed using one-way analysis of variance (Systat, Inc., Evanston, IL) with Tukey’s post-hoc test using R version 4.0.0. Asterisks represent the significant difference, as follows: * *p* < 0.05; ** *p* < 0.01; *** *p < 0*.001.

**Figure 6 ijms-21-04656-f006:**
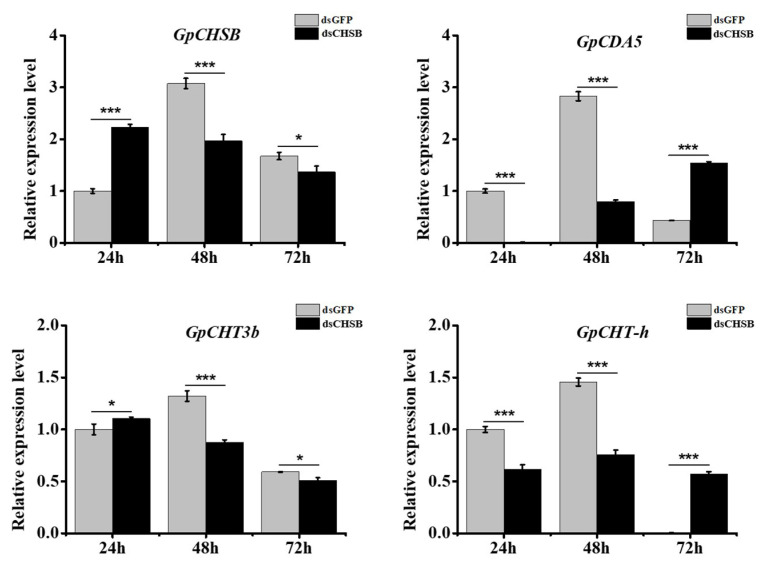
The analysis of expression patterns of *GpCHSB* and its downstream genes after injection with dsCHSB in the midgut at different times using RT-qPCR. *GpRpl32* was used to normalize the data that are showed as mean ±standard error, the mean is the triple independent repeats. The 2^−ΔΔ*C*t^ method was adopted to calculate the relative expression level. Differences among triple repeats were analyzed using one-way analysis of variance (Systat, Inc., Evanston, IL, USA) with Tukey’s post-hoc test using R version 4.0.0. Asterisks represent the significant difference, as follows: * *p < 0*.05; *** *p < 0*.001.

**Figure 7 ijms-21-04656-f007:**
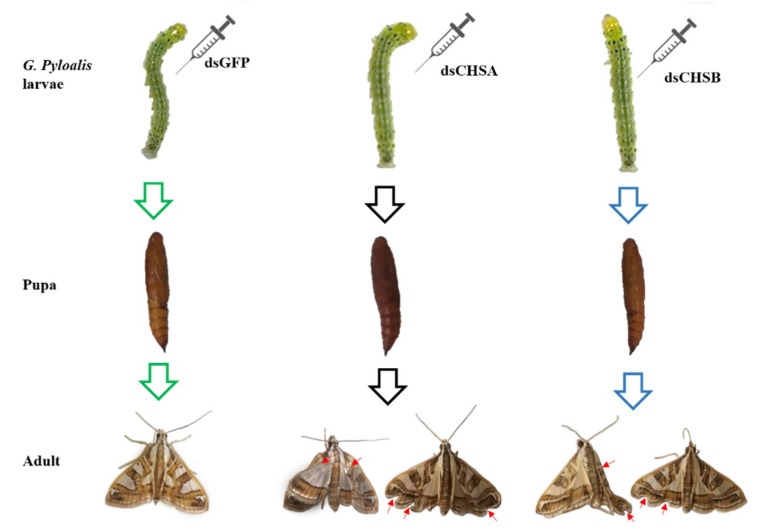
Representative phenotypes of *G. pyloalis* after dsRNA treatment. The phenotypes are taken after metamorphosis using Canon, PowerShot SX720.

**Figure 8 ijms-21-04656-f008:**
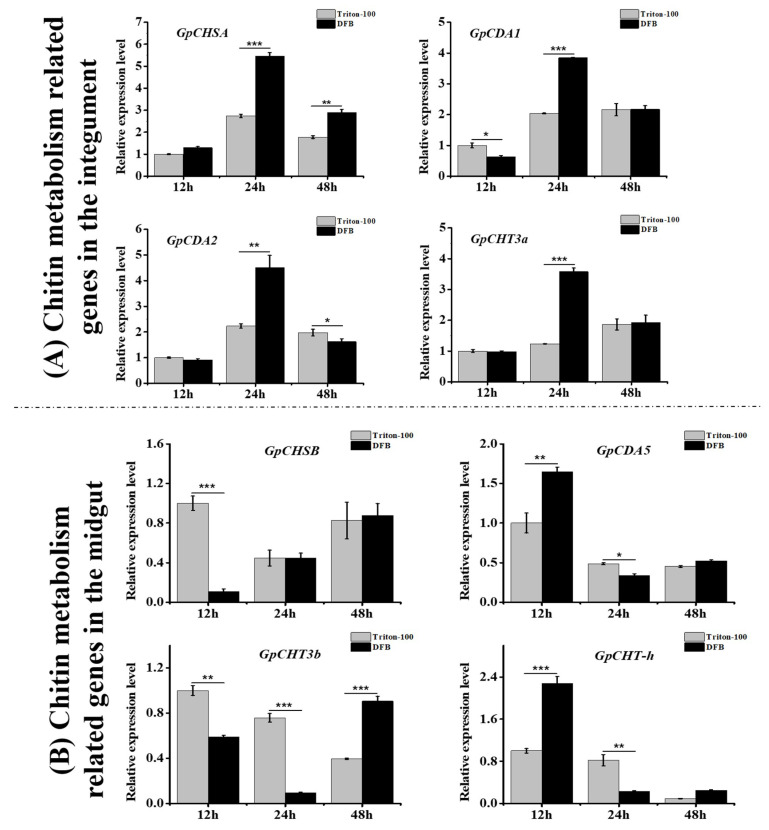
Assessment of expression levels of *GpCHSA*, *GpCHSB,* and their downstream genes post-DFB treatment. (**A**) Expression of *GpCHSA* and its downstream genes in the integument. (**B**) Expression of *GpCHSB* and its downstream genes in the midgut. *GpRpl32* was used to normalize the data that are showed as mean ±standard error, the mean is the triple independent repeats. The 2^−ΔΔ*C*t^ method was adopted to calculate the relative expression level. Differences among triple repeats were analyzed using one-way analysis of variance (Systat, Inc., Evanston, IL) with Tukey’s post-hoc test using R version 4.0.0. Asterisks represent the significant difference, as follows: * *p < 0*.05; ** *p < 0*.01; *** *p < 0*.001.

**Figure 9 ijms-21-04656-f009:**
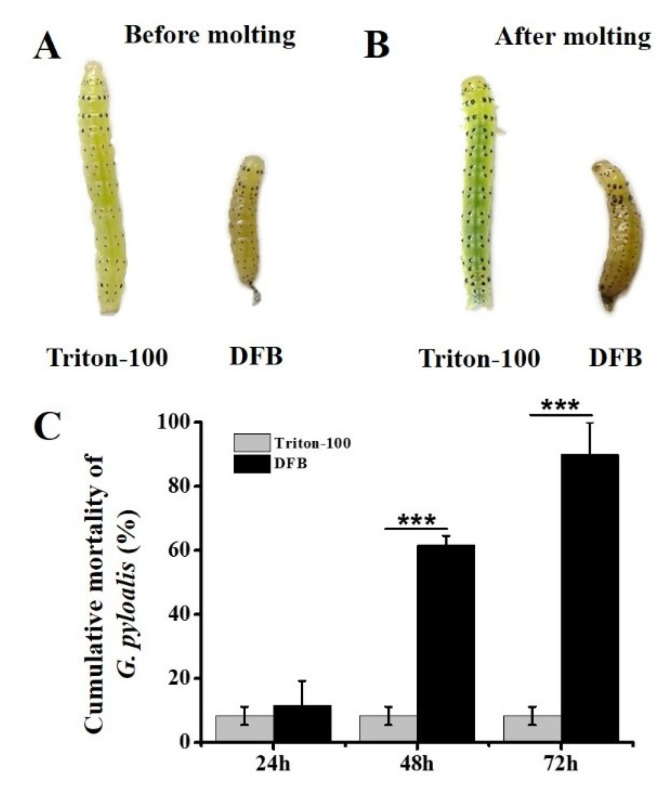
The effect of DFB on *G. pyloalis* development. (**A**-**B**) *G. pyloalis* phenotypes post-DFB and Triton-100 (control) treatment before and after molting. (**C**) Cumulative mortality of *G. pyloalis* after exposure to DFB and Triton-100. Differences among triple repeats were analyzed using one-way analysis of variance (Systat, Inc., Evanston, IL) with Tukey’s post-hoc test using R version 4.0.0. Asterisks represent the significant difference, as follows: *** *p < 0*.001.

**Figure 10 ijms-21-04656-f010:**
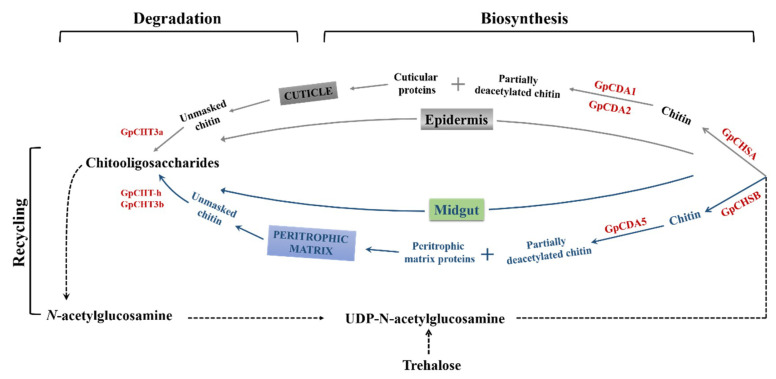
The hypothesis of the role of *GpCHSA* and *GpCHSB* and their downstream genes in chitin metabolism in *G. pyloalis*. *GpCHSA* and *GpCHSB* are involved in cuticle and PM chitin metabolism, respectively*. GpCHSA* regulates the expression of *GpCDA1*, *GpCDA2* and *GpCHT3a*. *GpCHSB* regulates expression of *GpCDA5*, *GpCHT-h* and *GpCHT3b*. Genes analyzed in this study were highlighted with the red color.

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
