# Peer review of "Identification, Characterization, and Functional Analysis of Chitin Synthase Genes in Glyphodes pyloalis Walker (Lepidoptera: Pyralidae)"

_ijms, 2020, doi:10.3390/ijms21134656_

Round 1
Reviewer 1 Report
In this MS, the authors identified two chitin synthase genes (GpCHSA and GpCHSB) from G. pyloalis. Their experimental results show that these two genes play a crucial role in the development and wing stretching in G. pyloalis adult. Their results reveal a high potential for pest control targets.
This paper is important for researchers working in a similar field. However, a major revision is required for getting acceptance of this paper. My main concern is the length. It is too long (15 pages, 10 figures, 2 tables). I would suggest switching some figures and at least one table to the Suppl info and reduce the text as possible without losing the important content.
Page 2, pages 48-49: “It is widely distributed throughout fungal cell walls, arthropod exoskeletons, and nematode eggshells [3, 4].” As the authors mentioned about chitin’s wide distribution, I would suggest including more references, especially marine invertebrates (see an example ref. below).
The authors mentioned (Page 2, line 48) that chitin is widely distributed throughout fungal cell walls. It is a wrong message for the readers. The chitin is not only containing in the fungal cell wall, but it is also contained in other cell walls (e.g., First evidence of chitin in calcified coralline algae: new insights into the calcification process of Clathromorphum compactum. Sci Rep 4, 6162 (2015). https://doi.org/10.1038/srep06162)
Author Response
Thank you for your professional suggestion, we have revised our manuscript after having carefully considered your comments, details are as follows: Major comments: 1. This paper is important for researchers working in a similar field. However, a major revision is required for getting acceptance of this paper. My main concern is the length. It is too long (15 pages, 10 figures, 2 tables). I would suggest switching some figures and at least one table to the Suppl info and reduce the text as possible without losing the important content. Reply: I have switched table 1 and table 2 to the supplementary material. 2. Page 2, pages 48-49: “It is widely distributed throughout fungal cell walls, arthropod exoskeletons, and nematode eggshells [3, 4].” As the authors mentioned about chitin’s wide distribution, I would suggest including more references, especially marine invertebrates (see an example ref. below). Reply: We have added marine organism into the manuscript, and related references you suggested was also inserted into the manuscript (Line 49). 3. The authors mentioned (Page 2, line 48) that chitin is widely distributed throughout fungal cell walls. It is a wrong message for the readers. The chitin is not only containing in the fungal cell wall, but it is also contained in other cell walls (e.g., First evidence of chitin in calcified coralline algae: new insights into the calcification process of Clathromorphum compactum. Sci Rep 4, 6162 (2015). https://doi.org/10.1038/srep06162) Reply: The description has been revised, and the reference you mentioned have been added into the manuscript (Line 49-50).Reviewer 2 Report
This is a review of the manuscript entitled “Identification, characterization and functional analysis of chitin synthase genes in Glyphodes pyloalis Walker (Lepidoptera: Pyralidae).” The basic goal is to examine the effect that chitin synthase suppression would have on insect development with the possible development of a new approach to manage G. pyloalis populations.
Line 40) The sentence beginning “In recent years …” does not work because it mixes several unrelated issues and confuses the timing of events.
- In recent years there have been severe losses to local mulberry farmers
- The improper use of pesticides has caused pyloalis outbreaks. I am sure that this is not recent behavior. At least in the USA improper use of pesticides has been, is, and will continue to be a problem.
- Climate change may result in additional stress and result in more damage to crops.
Line 44) What do you mean by abuse? Is this illegal use of pesticides (applying off-label)? Failure to rotate modes of action, or something else?
Line 51) The PM does not help the insect maintain its shape. Also what does “it” refer to in this sentence?
Line 66) replace comma with period, delete “and” and start a new sentence.
Line 6s7) add a citation. If it is “extensively recognized” then several citations or a review article. If it is just “recognized” then at least one citation.
Line 120) incomplete sentence.
Line 122) The sentence starting “A total of 26 CHSs were grouped …” does not work.
- Is the aqua CHSA or CHSB, is the pink CHSA or CHSB?
- The division seems wrong. The aqua is monophyletic. The pink is polyphyletic. That said, there is clearly a CHSA and a CHSB and pyloalis has both.
- I would remove Dm, Ag, Tc, and Lm from “pink” and leave them as white. It is possible that there are more CHSX, but it is hard to know what to make of this region without more data.
- There are 14 species listed. If each has a CHSA and CHSB then there should be 28 (not 26) entries in Figure 3.
- As a graphic showing the grouping of CHSA and CHSB this is a great figure. As an analysis of evolutionary relationships, it does not work as well. What you are saying is that the flies (Dm) and grasshoppers (Lm) are ancestral and this goes against every phylogenetic tree I have seen for the last 40 years.
- It is possible that the figure suffers from not having a proper outgroup to root the tree. The best would be a primitive insect, but a spider or crustacean, might work.
- While figure 3 is a phylogenetic tree, there are problems with equating that to “evolutionary relationships.”
- Within the stated objectives of this manuscript Figure 3 is appropriate and the analysis is good. Figure 3 has fatal problems if expanded to discuss “evolutionary relationships.” In place of species use “order” and look at the arrangement. It does not make sense. It starts with two flies (Ag, Dm), then a beetle (Tc), then a grasshopper (Lm), as basal groups then several lepidopterans with a grasshopper (Ms) in their middle. The tree is phylogenetically wrong. To look at phylogenetic relationships it needs a better outgroup, and it may need more species. Figure 3 could also be indicating a mistake in published sequences.
Line 133) The sentence is too long. Consider breaking it into several sentences.
Line 136) wrong verb tense.
Line 149) I know that R can do both analyses, and I strongly suspect that Systat can do both analyses. Why did you use both programs? This is not a problem; it is just strange.
Figure 4) I will assume that the expression levels in targets are for the adult only. Make this clear in the figure.
Line 171) levels were
Figure 7: include the instar that was injected. I haven't read the methods yet because that section comes after this section.
Lines 281-283) does not make sense.
Lines 309-310) What does the classification of life stages have to do with the number of samples?
Line 310) What age of larvae?
Line 310) Ten samples? Is this ten larvae, or the tissues from ten larvae?
Line 311) If ten samples were mixed, and there were 50 larvae, then there would be five replicates, but only three are reported, I do not understand.
Organize the methods in a logical order in which the steps would be completed. It was not “Rear insects’ then “bioinformatics analysis” followed by DNA extraction.
Line 364) Was that 30 larvae in the treatment and 30 for the control? How do these 30 larvae relate to the 3 biological replicates?
Author Response
Thank you for your professional suggestion, we have revised our manuscript after having carefully considered your comments, details are as follows:
Major comments:
- Line 40) The sentence beginning “In recent years …” does not work because it mixes several unrelated issues and confuses the timing of events. In recent years there have been severe losses to local mulberry farmers. The improper use of pesticides has caused pyloalis outbreaks. I am sure that this is not recent behavior. At least in the USA improper use of pesticides has been, is, and will continue to be a problem. Climate change may result in additional stress and result in more damage to crops.
Reply: Thank you for your professional comment. The inaccuracy description has been revised (Line 40-41).
- Line 44) What do you mean by abuse? Is this illegal use of pesticides (applying off-label)? Failure to rotate modes of action, or something else?
Reply: The use of pesticides is legal, the mean of abuse in the manuscript is the heavy use, and the abuse is replaced with heavy use (Line 43).
- Line 51) The PM does not help the insect maintain its shape. Also what does “it” refer to in this sentence?
Reply: We are sorry for the incorrect description. The mean of this sentence is the component of the cuticle plays an important role in keeping insect shape and peritrophic matrix (PM) plays an important role in protecting itself from external stresses, which has been revised (Line 51). The “it” refer to insect, which has been replaced with itself (Line 51).
- Line 66) replace comma with period, delete “and” and start a new sentence.
Reply: It has been revised according to your suggestion (Line 66).
- Line 67) add a citation. If it is “extensively recognized” then several citations or a review article. If it is just “recognized” then at least one citation.
Reply: Two citations including a review article have been added into line 67.
- Line 120) incomplete sentence.
Reply: It has been revised (Line 115-120).
- Line 122) The sentence starting “A total of 26 CHSs were grouped …” does not work.
Is the aqua CHSA or CHSB, is the pink CHSA or CHSB?
The division seems wrong. The aqua is monophyletic. The pink is polyphyletic. That said, there is clearly a CHSA and a CHSB and pyloalis has both. I would remove Dm, Ag, Tc, and Lm from “pink” and leave them as white. It is possible that there are more CHSX, but it is hard to know what to make of this region without more data.
There are 14 species listed. If each has a CHSA and CHSB then there should be 28 (not 26) entries in Figure 3.
As a graphic showing the grouping of CHSA and CHSB this is a great figure. As an analysis of evolutionary relationships, it does not work as well. What you are saying is that the flies (Dm) and grasshoppers (Lm) are ancestral and this goes against every phylogenetic tree I have seen for the last 40 years.
It is possible that the figure suffers from not having a proper outgroup to root the tree. The best would be a primitive insect, but a spider or crustacean, might work.
While figure 3 is a phylogenetic tree, there are problems with equating that to “evolutionary relationships.”
Within the stated objectives of this manuscript Figure 3 is appropriate and the analysis is good. Figure 3 has fatal problems if expanded to discuss “evolutionary relationships.” In place of species use “order” and look at the arrangement. It does not make sense. It starts with two flies (Ag, Dm), then a beetle (Tc), then a grasshopper (Lm), as basal groups then several lepidopterans with a grasshopper (Ms) in their middle. The tree is phylogenetically wrong. To look at phylogenetic relationships it needs a better outgroup, and it may need more species. Figure 3 could also be indicating a mistake in published sequences.
Reply: The phylogenetic tree has been regenerated according to your suggestions. Homologous genes in two outgroup species (Parasteatoda tepidariorum and Apis mellifera) and 3 more other species were added, and the inappropriate description in the manuscript has been revised (Line 126-132).
- Line 133) The sentence is too long. Consider breaking it into several sentences.
Reply: It has been revised (Line 141-144).
- Line 136) wrong verb tense.
Reply: It has been revised.
- Line 149) I know that R can do both analyses, and I strongly suspect that Systat can do both analyses. Why did you use both programs? This is not a problem; it is just strange.
Reply: Systat was used for drawing and analyzing variance in this study, and R version was used to analyze the significant difference.
- Figure 4) I will assume that the expression levels in targets are for the adult only. Make this clear in the figure.
Reply: Tissues used in this study were fifth instar larvae, which has been added into the legend (Line 156).
- Line 171) levels were
Reply: It has been revised.
- Figure 7: include the instar that was injected. I haven't read the methods yet because that section comes after this section.
Reply: The method was added into the manuscript (Line 387-388).
- Lines 281-283) does not make sense.
Reply: The description has been revised (Line 290).
- Lines 309-310) What does the classification of life stages have to do with the number of samples?
Reply: To detect expression levels of GpCHSA and GpCHSB in different life stages, ten samples were mixed to minimize individual genetic differences. The classification of life stages was for collecting 10 samples in different life stages. The description of this part has been revised (Line 335-337).
- Line 310) What age of larvae?
Reply: Fifth instar, and it has been added into line 338.
- Line 310) Ten samples? Is this ten larvae, or the tissues from ten larvae?
Line 311) If ten samples were mixed, and there were 50 larvae, then there would be five replicates, but only three are reported, I do not understand.
Reply: We are sorry for the unclear description. Ten samples were mixed to minimize individual genetic differences in detecting the relative expression level in different life stages. Different tissues of 50 larvae in fifth instar were dissected for detecting relative expression levels in different tissues. Triple replicates are the detection of each gene in these samples. The description of the method has been revised. (Line 335-340)
- Organize the methods in a logical order in which the steps would be completed. It was not “Rear insects’ then “bioinformatics analysis” followed by DNA extraction.
Reply: The methods have been organized again.
- Line 364) Was that 30 larvae in the treatment and 30 for the control? How do these 30 larvae relate to the 3 biological replicates?
Reply: Yes, 30 larvae in the treatment and 30 for the control. Thirty larvae is one repeat. The description have been revised (Line 383).
Round 2
Reviewer 1 Report
The authors nicely revised the paper so I have no any further issues to resolve.